# Two New Cases of Hypertrophic Cardiomyopathy and Skeletal Muscle Features Associated with *ALPK3* Homozygous and Compound Heterozygous Variants

**DOI:** 10.3390/genes11101201

**Published:** 2020-10-15

**Authors:** John Jorholt, Yulia Formicheva, Tatyana Vershinina, Artem Kiselev, Alexey Muravyev, Elena Demchenko, Petr Fedotov, Anna Zlotina, Anton Rygkov, Elena Vasichkina, Thomas Sejersen, Anna Kostareva

**Affiliations:** 1Department of Women’s and Children’s Health and Center for Molecular Medicine, Karolinska Institute, 17177 Stockholm, Sweden; thomas.sejersen@ki.se (T.S.); anna.kostareva@ki.se (A.K.); 2Almazov National Medical Research Centre, 197341 Saint Petersburg, Russiavershinina_tl@almazovcentre.ru (T.V.); myravyoval@mail.ru (A.M.); demchenko_ea@almazovcentre.ru (E.D.); fedotov_pa@almazovcentre.ru (P.F.); anna-zlotina@yandex.ru (A.Z.); ryzhkov_av@almazovcentre.ru (A.R.); vasichkina_ea@almazovcentre.ru (E.V.)

**Keywords:** hypertrophic cardiomyopathy, alpha kinase 3, *ALPK3*

## Abstract

Hypertrophic cardiomyopathy associated with damaging variants in the *ALPK3* gene is a fairly recent discovery, and only a small number of patients have been described thus far. Here we present two additional patients with hypertrophic cardiomyopathy caused by biallelic variants in *ALPK3*. Genetic investigation was performed using a targeted gene panel consisting of known cardiomyopathy-associated genes and whole exome sequencing. The patients showed a large difference in the age of onset, and both presented with extracardiac features that are often seen in *ALPK3* patients. The patient with the later onset showed milder extracardiac symptoms, such as decreased muscle tone and distal muscular dystrophy, but had fast progression of cardiac complications leading to the need of heart transplantation. This study further elucidates the variability of both symptoms and age of onset among these patients.

## 1. Introduction

Cardiomyopathies are a group of clinically heterogenous disorders that affect heart function and are the leading cause of cardiovascular morbidity and mortality among children and adults. Hypertrophic cardiomyopathy (HCM) is, together with dilated cardiomyopathy (DCM), the most common form of cardiomyopathy and the foremost cause of heart failure and sudden cardiac death [1]. It is characterized by increased ventricular wall thickness associated with diastolic dysfunction of ventricular chambers. The disorder is both genetically and phenotypically heterogenous with incomplete penetrance and variable expressivity. Various cardiac phenotypes can be caused by genetic variations in the same gene, and several genes can be the cause of the same type of cardiomyopathy [2]. Identifying the causative variation is, therefore, often challenging and time-consuming. The advancement of sequencing techniques, such as whole exome sequencing (WES), has led to the discovery of many new genes and variations associated with cardiomyopathies during the last decade. In particular, the vast proportion of causative genes of HCM have been uncovered, with most of the variants being localized within eight contractile sarcomeric genes [3]. However, there are still a number of HCM cases with rare or undefined genetic causes. This is particularly true for complex clinical phenotypes with a combination of cardiomyopathy and skeletal muscle involvement. Thus, standard guidelines including risk stratification, prognosis and treatment approaches are not validated for HCM associated with a neuromuscular phenotype or system metabolic disorders. Therefore, collecting and reporting clinical information for such rare forms of cardiomyopathies will facilitate building up the personalized system of treatment strategy and prognosis.

The alpha kinase 3 (*ALPK3*) gene was only recently described in connection to severe forms of pediatric cardiomyopathy, myopathic and dysmorphic skeletal features [4], while the mouse model of *Alpk3*-associated cardiomyopathy was described in 2012 [5]. *Alpk3* is specifically expressed in the cardiac crescent and developing heart in mouse embryos, as well as adult hearts and adult skeletal muscle [6]. It appears to play a critical role in cardiomyocyte differentiation. While detailed functions of this protein are not clearly described, it has been shown that *Alpk3* expression is induced during differentiation of mouse P19CL6 cells into cardiomyocytes prior to induction of cardiomyocyte-specific transcription factors. Overexpression of *Alpk3* enhanced cardiomyocyte differentiation, whereas knockdown of *Alpk3* expression inhibited cardiomyocyte differentiation in P19CL6 cells [6]. *ALPK3* may act as a transcriptional regulator of cardiomyocyte differentiation and heart development. *Alpk3*^−/−^ mice show gross cardiac enlargement with mural thickening typically associated with hypertrophic cardiomyopathy [5]. Cardiomyopathy in *Alpk3*^−/−^ mice is characterized by increased thickness of both left and right ventricular walls and reveals several features typically associated with dilated cardiomyopathy, namely, increases in end-diastolic and end-systolic volumes of the left ventricle, suggesting left ventricle chamber dilation. Using induced pluripotent stem cell approach (iPSC) from a consanguineous Pakistani family with two affected individuals, it was shown that *ALPK3*-deficient cardiomyocytes displayed abnormal calcium handling due to disorganized sarcomeres and an intercalated disc [7].

Hitherto, 27 patients with hypertrophic cardiomyopathy, caused by 19 different mutations in *ALPK3*, have been described [4,7,8,9,10,11] (Table 1). All but two of the described patients presented before the age of 18 and, apart from HCM, often revealed musculoskeletal abnormalities. The latter included scoliosis, knee and shoulder contractures, camptodactyly/arthrogryposis, axial hypotonia, short neck, dysmorphic facial features (low-set ears, high-arched palate, cleft palate and broad forehead) and pectus excavatum [8,9,11]. The majority of the described patients came from consanguineous families with varying degrees of relatedness (first- to sixth-degree cousins).

Here we present two new cases of HCM and skeletal muscle features associated with novel *ALPK3* variants. Detailed clinical features and genetic analysis allowed to broaden the phenotypic presentation of *ALPK3*-associated cases.

## 2. Materials and Methods

The study was performed according to the Declaration of Helsinki, and approval was obtained from Almazov National Medical Research Centre Ethical Committee. Written informed consent was obtained from the adult subject and the parents of the minor prior to investigation. Routine clinical examination included physical examination, 12-lead electrocardiography and Holter ECG monitoring, transthoracic echocardiography, MRI, neurological examination and biochemical and hormone tests.

### 2.1. Target and Whole Exome Sequencing

Target sequencing and whole exome sequencing were performed as previously described [12,13]. In short, DNA was purified from whole blood with a FlexiGene Kit according to the manufacturer’s recommendations, with an additional step of RNAase treatment (Qiagen, Venlo, The Neatherlands). For Patient 1, a targeted panel of 108 cardiomyopathy-associated genes (not including *ALPK3*) was initially studied using the Haloplex Target Enrichment System (Agilent, Waldbronn, Germany) with an Illumina MiSeq instrument (for gene list see Appendix A). After no pathogenic or likely pathogenic variants, or variants of unknown significance were identified in the genes with high expression levels in the myocardium, whole exome sequencing was performed using a SureSelect Human All Exon V6 r2 (60 Mbp) target enrichment kit (Agilent Technologies, Santa Clara, CA, USA) with an Illumina HiSeq instrument and SBSv4 chemistry (Illumina, San Diego, CA, USA).

For Patient 2, a targeted panel of 173 cardiomyopathy-associated genes (including *ALPK3*) was studied using the SureSelect Target Enrichment System (Agilent; Waldbronn, Germany) with an Illumina MiSeq instrument (see Appendix A).

Alignment, data processing and variant calling were performed according to GATK BestPractice recommendations (Broad Institute, Cambridge, MA, USA) using hg19 human genome reference. Variant annotation was performed using ANNOVAR (Philadelphia, PA, USA) [14].

For Patient 1, the list of variants from WES data was first filtered using expression data (GSE71613), keeping variants in genes with known expression in cardiac and skeletal muscle. Secondly, variants with frequency higher than 0.01% (GnomAD) [15] and deep intronic variants were filtered out. During the next step, variants were filtered based on their functional consequences, and the remaining variant types were nonsynonymous single-nucleotide variants, insertions, deletions and stopgain mutations. The remaining variants were filtered based on their predicted pathogenicity according to a number of prediction programs, such as CADD [16] and SIFT [17]. For Patient 1, the data were finally reduced using a list of 220 genes associated with cardiac phenotype variants (Cardio genes) (Appendix A).

In Patient 2, where a targeted panel was used, the variants were filtered in three steps. First, variants with minor allele frequency above 0.01% were discarded. Secondly, variants without functional consequences were removed, and lastly, variants with predicted pathogenicity were kept (Figure 1).

Assuming a recessive inheritance pattern of disease, homozygous and compound heterozygous variants would be prioritized from the lists of remaining variants of both patients after filtering. Bidirectional Sanger sequencing was performed to validate remaining variants, using polymerase chain reaction (PCR) primers covering the gene area of interest (Appendix A).

### 2.2. Cytogenetic Analysis

Peripheral lymphocyte karyotyping was performed according to standard protocol. Oligonucleotide array-based comparative genomic hybridization (array-CGH) was carried out using an Agilent 180K array with 13 kb median probe spacing (SurePrint G3 Human CGH Microarray, Agilent Technologies, Santa Clara, CA, USA). Clinical significance of the findings was evaluated based on data from DGV, OMIM and DECIPHER databases.

## 3. Results

### 3.1. Clinical Presentation

Patient 1 (Russian male, born 2008) was born to healthy parents after the fourth pregnancy (the first three pregnancies ended with medical abortion, pregnancy four—healthy boy, pregnancy five—miscarriage). The Apgar score was 7/8, birth weight—3.0 kg, birth length—50 cm. He was diagnosed with cardiac pathology due to severe cardiac hypertrophy at a routine prenatal ultrasound examination. His condition at birth was stable with a normal movement pattern and no signs of muscular hypotonia. There were no signs of progressive heart failure or respiratory distress. HCM diagnosis was confirmed at 6 months of age. During the first year of life, the patient demonstrated psychomotor delay, low weight gain, subluxation of hips and bilateral cryptorchidism. At 4 years of age, echocardiography revealed marked cardiac hypertrophy, left ventricular dysfunction with a slightly decreased ejection fraction (IVS 18 mm, *z*-score 5.9, EF 49%) and increased left ventricular (LV) trabeculation. The patient showed delay in psychomotor development, including speech and motor activity delay (started to walk independently at 1.5 years), inability to follow the school program and demonstrated dysmorphic features, such as broad forehead, cleft palate, axial hypotonia, short neck and pectus excavatum, as well as diffuse muscle wasting and scoliosis.

Echocardiography performed at the age of 7 years revealed marked asymmetric LV hypertrophy (IVS 24 mm, *z* score 6.9) which had progressed by 9 years of age (IVS 33 mm–*z* score 7.8) with no obstruction (Figure 2A,B). Cardiac MRI reported no chamber enlargement, slightly decreased left and right ventricular contractility and diffuse intramural late gadolinium enhancement compatible with HCM (Figure 2C–F). Conduction disturbances were not registered, apart from marked QT interval prolongation (480 ms, most likely due to QRS complex broadening) and pronounced repolarization abnormalities (Figure 2G,H). Repeated Holter monitoring revealed no episodes of ventricular tachycardia or premature ventricular beats, and no syncopes were reported. Cardiac enzymes remained elevated during the entire period (cardiac troponin I 0.238 ng/mL with reference interval 0.0000–0.0340, CK-MB 34.6 u/L with reference interval 0.0–24.0 and lactate dehydrogenase 279 u/L with reference interval 125–220). Creatine kinase levels remained normal at all test timepoints. At the age of 12, the patient had no progression of chamber enlargement or reduction of contractility, but died due to sudden cardiac death (SCD) after 30 min of unsuccessful resuscitation.

Patient 2 (Russian female, born 1986) was diagnosed with hypertrophic cardiomyopathy at the age of 21 years after a routine examination during pregnancy, but showed no clinical symptoms at the time. Two years later, frequent palpitations were noticed, together with dyspnea upon exercise, and heart failure symptoms progressed despite beta-blocker therapy. At that time, cardiac contractility remained preserved (EF 78%). She showed clinical deterioration two years later, after her second pregnancy and Caesarean section, when cardiac contractility reduction was registered (EF 38%) along with non-sustained ventricular tachycardia. Cardiac MRI revealed cardiac hypertrophy with no signs of obstruction (IVS 18 mm, PW 14 mm), decreased contractility of both ventricles (LV EF 40%, RV EF 40%), atrial enlargement and increased trabeculation of left ventricular posterior and lateral walls. No left ventricular outflow tract obstruction was registered. Morphological examination of myocardial biopsy material demonstrated prominent cardiomyocyte hypertrophy, disarray, signs of fibroelastosis and endocardium thickening, interstitial and perivascular fibrosis (Appendix A). The proportional area of cardiomyocytes corresponded to approximately 20%. Neurological examination revealed horizontal nystagmus, vertical gaze ophthalmoplegia, low muscle tone and distal muscle hypotrophy of the upper and lower limbs, muscle weakness of foot extensors (manual testing score of 3 on the manual muscle testing (MMT) scale 0–5), the toe extensor in particular, polyneuropathy and decreased sensitivity of distal limb areas (“gloves and socks”). However, electromyography did not confirm myopathic changes. Brain MRI showed bilateral periventricular white matter changes in the form of leucoaraiosis around the posterior corns of the lateral ventricles. The patient deteriorated over the following years with a decreased ejection fraction, severe left ventricular hypertrophy, angina pectoris and QT interval prolongation in ECG. At 30 years of age, she received implantable cardioverter defibrillator (ICD) implantation due to non-sustained ventricular tachycardia and syncopes. Starting at 33, she experienced several episodes of pulmonary edemas with need of inotropic support, NTproBNP was 2241 pg/mL. Cardiac respiration test documented a VO_2_ peak of 13.4 mL/kg/min. The patient was listed for heart transplantation, which was successfully performed at 33 years of age. No known family history of cardiac disease was reported, and both parents of the patient were healthy.

### 3.2. Sequencing and Genetic Analysis

Standard cytogenetic analysis of Patient 1 showed a normal male karyotype (46, XX), which excluded numerical or large-scale structural chromosomal rearrangements as a cause of the complex phenotype. To screen for copy number variants (CNVs), oligonucleotide array-based comparative genomic hybridization (array-CGH) was carried out and revealed no pathogenic variations. No cytogenetic analysis was performed for Patient 2.

Analysis of WES data, together with the list of 220 cardio-related genes, revealed a compound heterozygous mutation in *ALPK3* in Patient 1, consisting of two frameshift deletions in exon 5 (NM_020778): c.2033delG: p.R687fs and exon 6 (NM_020778): c.3558delG: p.V1186fs. None of the variants had previously been reported (Figure 3). No other variants of interest remained after all the filtering steps. The targeted exome had average coverage of 91.3 and 69.8% at read depths 10 × and 30 ×, respectively. Both variants were validated using Sanger sequencing. DNA samples from the parents and the brother of the patient were not available.

In Patient 2, target sequencing, using a panel of 173 cardiomyopathy-associated genes, combined with filtering and sorting of the variants, revealed a homozygous nonsynonymous single-nucleotide variant in exon 10 of *ALPK3* (NM_020778): c.G4897A: p.G1633R. The variant has been reported one time in heterozygous form in gnomAD (rs750258262), had a CADD score of 34, and was predicted to be damaging by SIFT (Figure 3). No other variants of interest remained after the three steps of filtering were performed. The panel had an average read depth of 206 × and average percentage coverage of 98.8% at 30 ×. The homozygous variant was validated using Sanger sequencing. DNA samples from the healthy parents of Patient 2 were not available.

## 4. Discussion

We reported two new cases of cardiomyopathies with skeletomuscular features associated with the *ALPK3* gene. Variations in this gene were reported to cause severe pediatric cardiomyopathy with a mixed phenotype of DCM, HCM and left ventricular non-compaction (LVNC) [4,10,11]. Notably, in spite of an unfavorable course of disease in the first days and months, many surviving children demonstrated marked improvement of cardiac function with age. Several reported cases were characterized by a transition from DCM to HCM, a unique phenotypic phenomenon, described only for *ALPK3*-associated cases so far [10,11]. In spite of the severe hypertrophic phenotype, no obstruction has been reported for *ALPK3*-associated cases. In both cases presented here, we did not observe a transition from the DCM to the HCM phenotype. In contrast, both patients presented with severe HCM with no obstruction. In both cases, cardiac contractility decreased without marked cardiac dilation, but led to the need of cardiac transplantation in the second case. Remarkably, the latter was the first case of symptomatic *ALPK3*-associated cardiomyopathy presenting in adulthood and leading to cardiac transplantation 10 years after the presentation of symptoms. Two previously reported cases of adult *ALPK3*-associated cardiomyopathy described by Al Senaidi and co-authors were detected using cascade family screening and presented no clinical symptoms by the third and fourth decades of life, in spite of homozygote status [10]. Together with our two cases, this further elucidated the heterogeneity of *ALPK3*-related disorders with high variability in age of presentation, cause of the disease, transformation of clinical phenotypes and prognosis. This further includes the variability of the skeletal and muscle system involvement in *ALPK3*-associated cardiomyopathy. Musculoskeletal abnormalities were not reported in all patients with pathogenic *ALPK3* variants, and examples of reported abnormalities included subtle signs of muscle weakness, dysmorphic features and skeletal abnormalities [8,9]. In the presented cases, both patients revealed decreased muscle tone, axial weakness and clinical signs of polyneuropathy, but no defined changes were detected in Patient 2 using neurography and electromyography. Together with other reported cases we further stressed the importance of skeletal and neuromuscular system examination in patients with *ALPK3*-associated cardiomyopathy, especially in view of possible need of cardiac transplantation in the future. Therefore, the detailed degree of skeletal and muscle system involvement, as well as intellectual and endocrine status in *ALPK3*- associated cardiomyopathy cases, need to be further delineated.

In both presented cases, ventricular arrhythmias led to deterioration of the clinical phenotype. In Patient 2, this resulted in ICD implantation according to HCM risk stratification criteria and allowed survival until heart transplantation. However, in Patient 1, no signs of ventricular rhythm disturbances were detected, in spite of yearly routine checks and Holter monitoring, and no syncopes were reported. Therefore, SCD was the first and only presentation of cardiac arrhythmic disease for this patient. Together with data reported by Almomani and co-authors on early episodes of ventricular fibrillation in a homozygote carrier of *ALPK3*, our data stressed the importance of searching for informative ICD implantation criteria in pediatric patients with atypical forms of HCM, including *ALPK3*-associated cardiomyopathies [4].

The difference in severity of the phenotypes could be related to the variations’ different coding consequences and how they affect protein function and expression. However, a recent study by Herkert et al. showed no association between variant type or location, disease severity and/or extracardiac manifestations [11]. This is only the second report of a homozygous nonsynonymous variation in *ALPK3* associated with a cardiomyopathic phenotype. The patient with a glycine to arginine substitution in exon 10 showed later onset of symptoms, compared to the majority of previously reported cases, but had more severe progression, leading to the need of heart transplantation. It could, therefore, be recommended that younger asymptomatic carriers have regular check-ups to detect signs of deterioration. 

In conclusion, the two patients in our study further showed the variability of symptoms and onset of disease that is seen in cases of cardiomyopathy caused by variations in the *ALPK3* gene.

## Figures and Tables

**Figure 1 genes-11-01201-f001:**
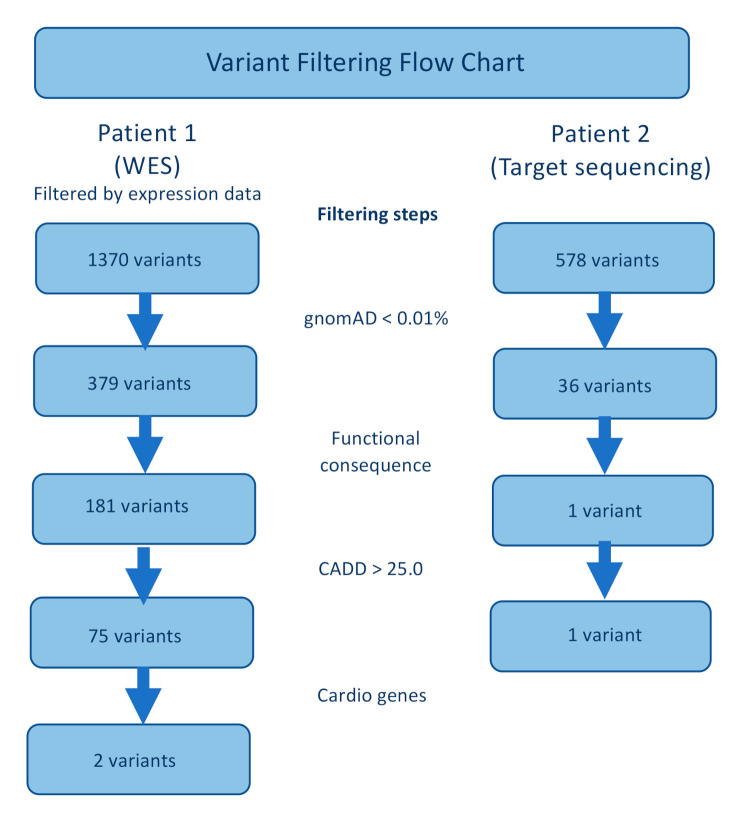
Genotyping flowchart for Patients 1 and 2.

**Figure 2 genes-11-01201-f002:**
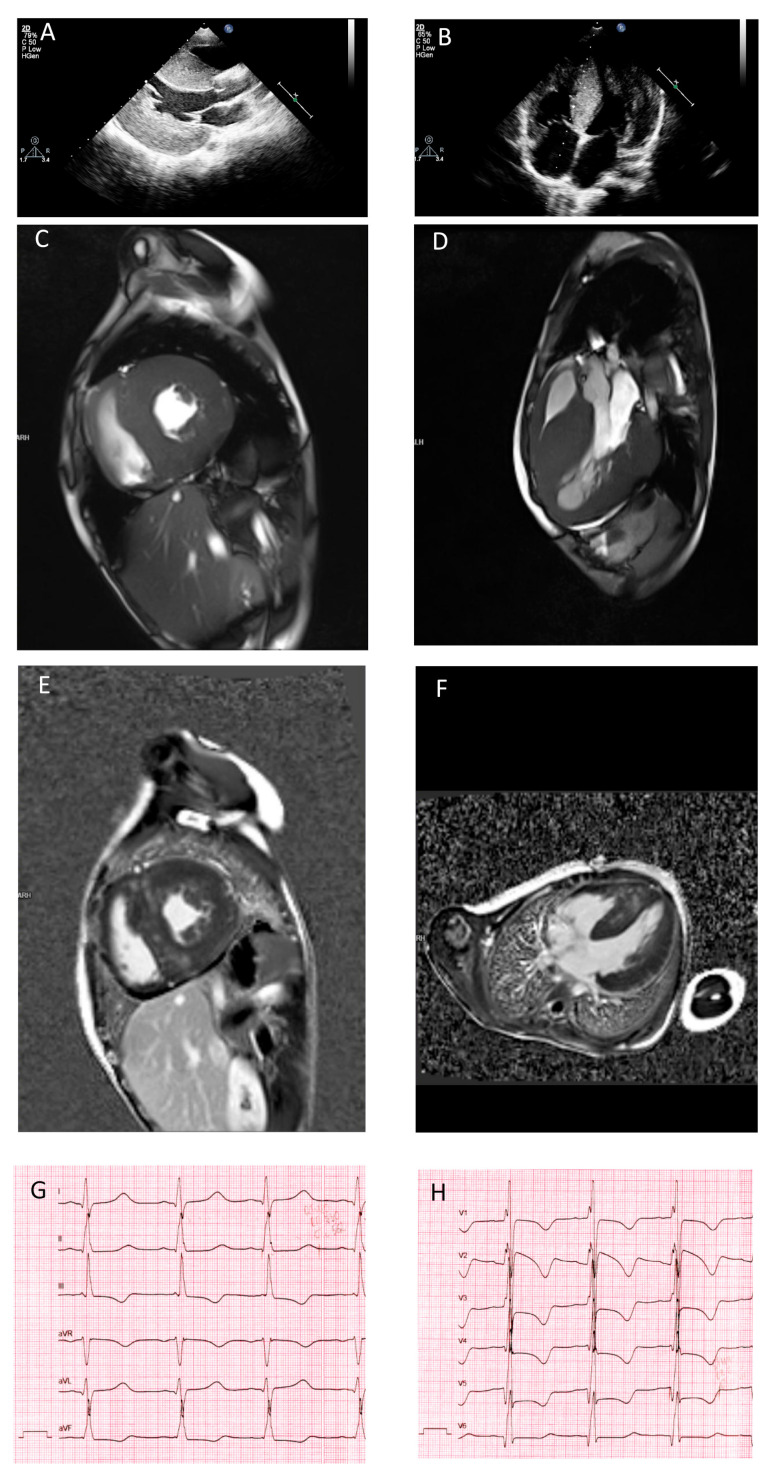
Morphological features of cardiomyopathy linked to *ALPK3* variants. Echocardiography picture of Patient 1 corresponding to long-axis view (**A**) and four-chamber view (**B**) confirmed severe symmetrical wall hypertrophy and atrial enlargement. Cardiac MRI images of Patient 1 in the short (**C**) and long (**D**) axes, demonstrating wall hypertrophy, limited cavity volume and late gadolinium enhancement (**E**,**F**). Repolarization abnormalities of Patient 1 detected by ECG (**G**,**H**).

**Figure 3 genes-11-01201-f003:**
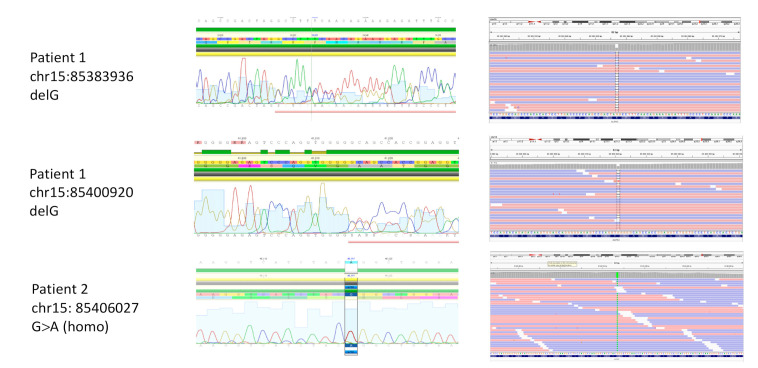
*ALPK3* variants, leading to cardiomyopathy in combination with skeletal muscle features. Sequencing data obtained using whole exome (Patient 1) and target (Patient 2) sequencing confirmed by Sanger sequencing. Compound heterozygous variants c.2033delG: p.R687fs and c.3558delG: p.V1186fs in Patients 1, and homozygous variant c.G4897A: p.G1633R in Patient 2.

**Table 1 genes-11-01201-t001:** Summary of *ALPK3* mutations resulting in hypertrophic cardiomyopathy.

Ref	Alt	Genotype	Variant Type	Amino Acid Change	Reference
G	A	Hom	Transition (acceptor site)	c.4736-1G > A	Almomani et al. [4]
C	T	Hom	Stopgain	c.3781C > T: p.R1261X	Almomani et al. [4]
G	A	Hom	Stopgain	c.5294G > A: p.W1765X	Almomani et al. [4]
G	A	Hom	Stopgain	c.3792G > A: p.W1264X	Phelan et al. [7]
C	-	Hom	Frameshift deletion	c.2018delC: p.Q675SfsX30	Çağlayan et al. [8]
AA	-	Hom	Frameshift deletion	c.1531_1532delAA: p.K511RfsX12	Jaouadi et al. [9]
G	A	Hom	Stopgain	c.639G > A: p.W213X	Al Senaidi et al. [10]
C	T	Comp het	Stopgian	c.1018C > T: p.Q340X	Herkert et al. [11]
G	A	Comp het	Nonsynonymous SNV	c.2434G > A: p.V812M	Herkert et al. [11]
C	-	Comp het	Frameshift deletion	c.4332delC: p.K1445RfsX29	Herkert et al. [11]
G	-	Comp het	Framehsift deletion	c.541delG: p.A181PfsX130	Herkert et al. [11]
C	T	Comp het	Nonsynonymous SNV	c.3439C > T: p.R1147W	Herkert et al. [11]
A	-	Comp het	Frameshift deletion	c.4997delA: p.N1666TfsX14	Herkert et al. [11]
G	C	Comp het	Nonsynonymous SNV	c.4091G > C: p.G1364A	Herkert et al. [11]
G	C	Comp het	Transition (donor site)	c.5105 + 5G>C	Herkert et al. [11]
G	T	Comp het	Nonsynonymous SNV	c.597G > T: p.E199D	Herkert et al. [11]
G	T	Comp het	Nonsynonymous SNV	c.4888G > T: p.V1630F	Herkert et al. [11]
C	T	Hom	Transition (donor site)	c.3418C > T: p.Q1140X	Herkert et al. [11]
G	C	Hom	Nonsynonymous SNV	c.5155G > C: p.A1719P	Herkert et al. [11]
G	-	Comp het	Frameshift deletion	c.2033delG: p.R687fs	This study
G	-	Comp het	Frameshift deletion	c.3558delG: p.V1186fs	This study
G	A	Hom	Nonsynonymous SNV	c.G4897A: p.G1633R	This study

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
