# Peer review of "Two New Cases of Hypertrophic Cardiomyopathy and Skeletal Muscle Features Associated with ALPK3 Homozygous and Compound Heterozygous Variants"

_genes, 2020, doi:10.3390/genes11101201_

Round 1

Reviewer 1 Report

Damaging variants (predominantly homozygous) in the ALPK3 gene have been identified as a rare cause of hypertrophic cardiomyopathy (HCM). This manuscript describes two additional cases with biallelic variants in ALPK3. Whilst the association with disease is not novel, as very few cases have been described this work adds to the phenotypic and genotypic information for this form of HCM.

Minor comments:

  1. It would be helpful to include details of ethnicity,
  2. were either of the two patients described from consanguineous families? I note the methods describe that a recessive inheritance pattern was prioritised
  3. Was genotyping carried out in the parents? are there any other siblings for patient 2? pedigrees would be helpful for both described cases.
  4. Can you include the myocardial biopsy images for Patient 2

Author Response

  1. Information about ethnicity was added to the clinical presentations of both patients. They are both of Russian decent.
  2. None of the patients come from consanguineous families. No other family members were affected and therefore the disease are assumed to be of a recessive nature.
  3. DNA samples were unfortunately not available for any family members. Patient 1 has healthy parentes and one healthy sibling. Patient 2 has healthy parents and no siblings.
  4. Pictures of histological staining of the myocardial biopsy has been added to the supplementary materials (Supplementary Figure 1).

Reviewer 2 Report

This study describes two HCM patients with ALPK3 variants, a rare recessive cause of this condition that has been previously described in a small number of reports. It provides detailed clinical data from the two patients (although very limited data on other family members), further enhancing our knowledge about the heterogeneous phenotypes associated with pathogenic variants in this gene.

My main concerns about this manuscript concern the clarity of the variant filtering procedure for the two patients and the lack of consistency between the methods text and Figure 1.
- The text describes filtering by expression in cardiac/skeletal muscle but this is not presented in the figure.
- The figure mentions "functional consequence" and the text "filtering by type" without any detailed description of what these entail.
- For WES, data for 220 cardiac genes were assessed - was this before the filtering steps in the figure?

As patient 1 presented with a recessive phenotype, the most powerful filtering step is to assess homozygous or compound heterozygous variants - this is mentioned in the text but it is not clear in the figure where this is applied. Were there any other candidates for recessive inheritance (either within the 220 selected genes or for rare protein altering variants in other genes)?

Overall, the variant filtering methods and results should be rewritten and clarified.

For patient 1, was the genotype status for the parents checked to ensure they were both heterozygous carriers of one ALPK3 truncating variant? The authors state they were healthy - was any cardiac phenotyping performed?

For patient 2, were her parents consanguineous? Were any other family members assessed either genetically or clinically? Most ALPK3 variants detected thus far have been truncating variants (usually biallelic), so the detection of a homozygous missense variant is relatively novel. Were any other variants detected in the panel sequencing that could contribute to the HCM phenotype?

Minor point:
Type ("filtrering") in Figure 1.

Author Response

The filtering steps have been clarified in both the text (rows 103-120) and figure 1.

The filtering by gene expression data was performed for the WES data as a first step before the rest of the filtering procedures.

The filtering based on the functional consequences caused by the variants has been clarified in the text.

A clarification about when the list of cardio genes was used was also added.

The assessment of the variants and their inheritance patterns was conducted after the filtering steps shown in Figure 1.

No other interesting variants were identified in any of the patients and information about that was added to the results.

DNA samples from the parents of patient 1 was not available and no family history of cardiac disease was reported.

The parents of patient 2 are not related and DNA samples were not available. There are not reports of a family history of cardiac disease. No other possibly contributing variants were found using the gene panel.